# Malignant Germ Cell Tumors and Their Precursor Gonadal Lesions in Patients with XY-DSD: A Case Series and Review of the Literature

**DOI:** 10.3390/ijerph18115648

**Published:** 2021-05-25

**Authors:** Sahra Steinmacher, Sara Y. Brucker, Andrina Kölle, Bernhard Krämer, Dorit Schöller, Katharina Rall

**Affiliations:** Department of Women’s Health, Tübingen University Hospital, Calwerstr. 7, 72076 Tübingen, Germany; Sahra.Steinmacher@med.uni-tuebingen.de (S.S.); sara.brucker@med.uni-tuebingen.de (S.Y.B.); Andrina.koelle@med.uni-tuebingen.de (A.K.); bernhard.kraemer@med.uni-tuebingen.de (B.K.); dorit.schoeller@med.uni-tuebingen.de (D.S.)

**Keywords:** XY-DSD, gonadectomy, malignant transformation

## Abstract

The risk of gonadal germ cell tumors is increased over the lifetime of patients with XY-disorders of sex development (XY-DSD). The aim of this study was to evaluate clinical features and histopathological outcome after gonadectomy in patients with XY-DSD to assess the risk of malignant transformation to gonadal germ cell tumors. Thirty-five women treated for XY-DSD at our hospital between 2003 and 2020 were enrolled in this study. Twenty-seven (77%) underwent prophylactic gonadectomy, 10 (29%) at our department and 17 (48%) at external hospitals. Eight (23%) patients didn’t receive gonadectomy. Of the patients who underwent a surgical procedure at our hospital, two patients were diagnosed with a unilateral seminoma, one patient with a bilateral and one patient with a unilateral Sertoli cell adenoma. According to these findings, preventive gonadectomy in patients with XY-DSD should be taken into consideration. Guidelines concerning the necessity of gonadectomy to avoid malignant transformation are still lacking. The risk of malignant germ cell tumors from rudimentary gonads has not been investigated sufficiently to date, as it is mostly based on case series due to the rarity of the condition. In our study we retrospectively analyzed patients who partly underwent bilateral gonadectomy, aiming to fill this gap. Concerning the ideal point of time for gonadectomy, further studies with a higher number of patients are needed.

## 1. Introduction

Disorders of sex development (DSD) are defined as congenital conditions with atypical gonadal, chromosomal, or anatomical sex. They usually present with atypical genitalia in the newborn or delayed puberty in the adolescent period [1].

According to the Chicago classification of DSD, patients are classified into the categories sex chromosome DSD (Klinefelter or Turner Syndrome, chimerism), XX-DSD, and 46 XY-DSD (disorders of testicular development and disorders of androgen synthesis or action) [2]. DSD affects 1 in 4500–5000 live births, mostly due to genetic defects during sexual differentiation [3]. XY-DSD in particular affects 1 in 20,000 births [4].

Variants of DSD patients with female phenotype and Y chromosome include 46 XY pure gonadal dysgenesis, 46 XY partial gonadal dysgenesis, androgen insensitivity syndrome (complete or partial) or 46, XY 17 alpha hydroxylase/17, 20-lyase deficiency among others. 

In pure gonadal dysgenesis, normal gonadal development is inhibited and patients are phenotypically female [5]. In partial dysgenesis, testis determination is incomplete, leading to a phenotype which depends on the degree of gonadal function [5].

Androgen insensitivity syndrome is based on the mutation of a gene coding for the androgen receptor. This X-linked recessive disease leads to insensitivity for circulating androgens, resulting in chromosomal and gonadal male individuals, but who are phenotypically females [6]. In complete androgen insensitivity syndrome (cAIS), mutations of the androgen receptor (AR) can be seen in 95% of all patients, whereas in partial androgen insensitivity syndrome it can be detected in up to 50%. Diagnosis is verified through identifications of mutations in the AR gene [7].

In the management of patients with XY-DSD, the increased risk of invasive germ cell tumors compared to the general population has to be taken into consideration, as the presence of Y chromosomal material serves as a risk factor for malignant transformation [5,8]. Some patients are at higher risk of developing pre-invasive tumors such as germ cell neoplasia in situ (GCNIS) or gonadoblastoma, which is commonly observed in complete or partial dysgenesis [9,10]. Invasive malignant germ cell tumors that can occur are seminoma, non-seminoma, or a combination of both [11].

Risk factors for malignant transformation include cryptorchism and presence of certain gene sequences [12]. The risk for gonadal tumor is lowest in patients with cAIS, while it can be up to 60% in patients with 46, XY gonadal dysgenesis [12].

In the past, bilateral gonadectomy was usually performed in childhood due to the reported risk of malignant transformation of gonads [13]. However, data proving a low potential of malignant transformation during childhood in cAIS has led to a more testis sparing management [14].

The risk of malignant transformation in cAIS increases with age and rarely occurs before puberty [15,16].

This testis sparing approach allows a normal puberty development as endogenous androgens are converted to estrogens through peripheral aromatization without the need for hormonal replacement therapy [14]. Yet in rare conditions, such as Leydig cell hypoplasia or 5a-reductase deficiency, gonadectomy is still often performed at an early age, so data regarding the prevalence of gonadal tumors is limited [12].

New findings underline a more individualized approach towards gonadectomy depending on localization of the gonads, internal and external phenotype and sex of rearing [17]. The risk of malignant transformation is highest in abdominal localization [17].

About 15% of women refuse the removal of their gonads in adulthood [18].

Early detection of a malignant or pre-malignant tumor is always important, but in this case difficult, because the limitation of tumor markers and US or MRI/CT to detect them and missing guidelines for surveillance. As no data exist concerning the risk of malignant transformation of gonads in adulthood for patients with XY-DSD, the treatment of these patients is challenging for the responsible medical staff [19,20].

Therapeutic considerations for gonadectomy regarding the potential risk of malignant transformation are derived from a case series of patients who underwent gonadectomy in adolescence due to the rarity of the disease [19,21,22]. Controversies exist regarding tumor risk, surgical timing, and postoperative management for the different variants [23]. Based on current findings, gonadal tumor risk is estimated according to molecular diagnosis of the patient [5]. To date, no management guidelines concerning the need and optimal point of time for gonadectomy in postpubertal individuals with XY-DSD are available.

In our study we retrospectively analyzed patients treated for XY-DSD at our department and who partly underwent bilateral gonadectomy in order to create data that can possibly be integrated into guidelines for the treatment of XY-DSD.

## 2. Materials and Methods

Women treated for XY-DSD at the Department of Women’s Health at the University Hospital Tuebingen between 2003 and 2020 and who possibly underwent bilateral gonadectomy were enrolled in this retrospective analysis. In case of gonadectomy, a detailed pathology analysis was performed. Patients usually presented with the request for neovagina to be created, which was performed using the laparoscopically assisted modified Vecchietti technique [24,25].

During the first visit the patients were evaluated for any family history. A general physical examination targeting the presence of a uterus and gonads, breast development and external genitalia, ultrasound, hormonal levels, and karyotype analysis was performed.

The general characteristics of the patients included female phenotype, karyotype with Y chromosome, and primary amenorrhea. Diagnosis was based on molecular genetic analysis, when possible.

Data extraction was performed from the clinical records. The histological criteria according to WHO were applied. Approval by the institutional ethics committee of the Medical Faculty of the Eberhard-Karls-University was obtained (506/2020BO).

Baseline characteristics were described by mean and standard deviation (SD). Basic statistics were performed using Excel version 16.27.

## 3. Results

A total of 35 patients with XY-DSD were included. Twenty-seven patients (71%) were diagnosed with complete androgen insensitivity syndrome, and four (9%) with partial androgen insensitivity syndrome. One patient (3%) was diagnosed with 5a-reductase deficiency, one (3%) with homozygous LD-Receptor deficiency, one (3%) with 17-ß-hydroxysteroid dehydrogenase deficiency and one patient (3%) with XY-DSD, not further specified. In 15 patients (43%), the diagnosis was confirmed by molecular genetic analysis (see Table 1).

### Prophylactic Gonadectomy

Twenty-seven (77%) of the patients underwent prophylactic gonadectomy, 10 (29%) at our department and 17(48%) in external hospitals. Of these, only one patient underwent unilateral gonadectomy. Eight patients (23%) did not receive a gonadectomy. Of the patients who underwent a surgical procedure at our hospital, two were diagnosed with unilateral seminoma (20%), one with a bilateral and one with a unilateral Sertoli-cell-adenoma (20%). Their history is presented below. Histopathological findings were not always available for patients who underwent gonadectomy at external hospitals. In these cases, histology was assumed to be benign if not otherwise reported. For patient characteristics see Table 1. The mean age of all patients at gonadectomy was 14.5 years (SD 9.27 years).
**Patient 1**: The patient presented to our department at the age of 20 years requesting creation of a neovagina. She had been diagnosed with cAIS in childhood; diagnosis was based on karyotyping and family history, as one sister, one aunt and two cousins had been diagnosed with cAIS. On clinical examination the patient displayed normal external genitalia, aplasia of the uterus and a blind ending vagina. The patient had a history of surgery for an inguinal hernia. The patient also presented with a neurogenic voiding disorder of unknown origin. The gonads had been previously monitored sonographically at another hospital and were removed during laparoscopic neovagina creation as requested by the patient after informed consent. The histopathological findings revealed a bilateral Sertoli-cell-adenoma. Hormonal substitution was prescribed with an oral contraceptive as preferred by the patient.**Patient 2**: The patient presented at the age of 20 years with vaginal and uterus aplasia and the request for neovagina creation. She had been referred to a gynecologist initially due to primary amenorrhea. cAIS was diagnosed and confirmed based on karyotype and molecular genetic analysis with deletion of Exon 1 of the androgen receptor gene. The patient presented with hypotrophic mammae, and scanty pubic and missing axillary hair. Serum testosterone levels were elevated. The patient underwent laparoscopic neovagina creation and bilateral gonadectomy. Bilateral Sertoli cell adenoma was found on the histopathological examination. The patient was prescribed hormonal replacement therapy.**Patient 3**: The 40-year-old patient sought consultation regarding gonadectomy for previously diagnosed cAIS. Tissue samples had been previously taken laparoscopically with benign histology. On ultrasound examination the left gonad was enlarged to 4 × 2 × 2 cm, confirming the indication for laparoscopic gonadectomy. The histopathological findings revealed a seminoma and a Leydig cell tumor in the left gonad. Tumor markers were negative and a CT-scan did not reveal any evidence for metastasis. The interdisciplinary tumor conference then agreed on active surveillance with clinical and ultrasound examination every three months, as well as MRI of the abdomen and chest CT every 6 months for two years. Hormonal replacement therapy was prescribed.**Patient 4**: The 24-year-old patient presented at our department for neovagina creation. Karyotype was XY and there was clinical evidence of 5α-reductase deficiency. The preoperative ultrasound did not show any suspicious findings. After laparoscopy with neovagina creation and bilateral gonadectomy, a seminoma was diagnosed in the right gonad. Medical aftercare consisted of clinical examination and imaging every three months.

A review of the literature on the frequency of germ cell tumors is given in Table 2.

## 4. Discussion

In our study we retrospectively analyzed patients treated for XY-DSD at our department and who partly underwent bilateral gonadectomy. Of the 10 patients who received gonadectomy at our clinic, three patients with cAIS and one patient with 5α-Reductase deficiency were diagnosed with abnormal histopathological findings. A seminoma was diagnosed in a 23-year old patient with 5 α-Reductase deficiency. A 40-year old patient with cAIS was diagnosed with Leydig-cell tumor and seminoma.

In a 19-year old patient with cAIS, a bilateral Sertoli cell adenoma was found, and in a 20-year old patient with cAIS, a unilateral Sertoli cell adenoma was diagnosed. Whereas seminomas are derived from testicular germ cells, Leydig cell or Sertoli cell tumors originate from testicular stroma cells [13,32,33]. Sertoli cell adenomas are almost always benign, even though cases of Sertoli cell adenocarcinoma have been described, usually occurring in adult life [34]. In our patient collective histopathological findings were mainly observed in patients with cAIS who received gonadectomy after adolescence.

Generally, and in contrast to our results, the risk for tumor development is considered to be lower in cAIS than in other XY-DSD. Hannema et al. demonstrate an incidence of two cases of carcinoma in situ in also postpubescent patients (4.5%) out of 44 gonadectomized patients with cAIS [26].

Cools et al. postulated that a rapid loss of germ cells in cAIS happens from one year on, serving as a factor that might modify the risk for germ cell tumors [35], whereas up to two thirds of germ cells are maintained in pAIS [36]. Thus, patients with cAIS and a higher apoptosis rate of germ cells in early life might also have a lower risk for tumor development [26,36]. Gonads in patients with cAIS usually also display more normally differentiated testicular tissue [37].

The risk of malignant transformation might also be influenced by responsiveness of the gonads to androgens and their hormonal function, [35] as prevalence of germ cell neoplasia is lower in patients with cAIS (5–7%), than in pAIS (15–20%) [26,29,36].

Similar findings to our study were demonstrated by Liu et al. in their investigation of 102 XY-DSD patients with a mean age of 20.7 years. Here, patients with cAIS had the highest risk for malignant transformation of gonads (30%) compared to patients with pAIS with 16.7%. Yet, this could be also due to the rather old patient collective [31], as the risk of gonadal malignant transformation is lower in patients with AIS before puberty compared to patients with other DSD [16] and this risk might increase after adolescence [7,13]. In our patient cohort mean age of all patients at gonadectomy was 14.5 years; in the group of four patients with relevant pathological findings it was 25.5 years. This underlines the fact that the risk for tumor development increases after adolescence.

Comparing other XY-DSD, Slowikowska-Hilczer et al. showed that the frequency of germ cell tumors is higher in DSD-patients with bilateral dysgenetic testes, than in patients with an asymmetry in gonadal dysgenesis [30]. In their study examining the frequency of the benign gonadoblastoma, Cools et al. found a gonadoblastoma in 10 of in total 14 (71.4%) gonadectomy samples in 12 patients with XY-DSD. Malignant dysgerminoma was found in one gonad (7.1%). Both gonadoblastoma and dysgerminoma were found in two gonads (14.2%) [27]. Beaulieu et al. examined the gonads of 30 patients, 22 with pure and eight with mixed gonadal dysgenesis. Of the totally 51 gonads, nine (18%) revealed tumor tissue [28]. By evaluating gonads laparoscopically in 40 patients with XY-DSD, Wünsch et al. performed gonadectomy in 19 patients. Tumors were diagnosed in four patients, three of them with complete gonadal dysgenesis. In 21 patients, biopsies were taken with no suspicious results [29]. These findings of an increased risk for malignant transformation in patients with complete gonadal dysgenesis are in line with the results of the study of Huang et al. Here a patient cohort of 292 patients with XY-DSD and gonadectomy was evaluated. Of the 59 patients with complete gonadal dysgenesis, germ cell tumor was diagnosed in 21 (35.5%). In contrast to our study, risk in androgen insensitivity syndrome was comparably lower. In 15 of 113 patients with AIS (mixed complete and partial) a neoplasia was found (13.2%) [23].

In our study, only one patient received gonadectomy due to suspicious findings in an ultrasound screening. The other patients didn’t show any preoperative abnormalities in the imaging.

Nakahl et al. examined MR images and histological correlation in 25 patients with cAIS. Of these, 13 eventually received gonadectomy and premalignant lesions were diagnosed in four patients (31%). Whereas Sertoli cell adenomas, which possibly harbor premalignant lesions, could be diagnosed in more than 80%, none of the histologically diagnosed premalignancies had previously been detected in the MRI [19]. This emphasizes the need for gonadectomy in cAIS and other XY- DSD as imaging does not represent a sufficient instrument for monitoring gonads [19]. For patients with cAIS and remaining testes, no reliable diagnostics exist.

To date there are no serum markers, such as ß-HCG or alpha-fetoprotein, that might detect lesions early [38]. MicroRNA clusters have been associated with invasive forms and germ cell tumors both in DSD and male patients. They might be promising in the diagnosis and follow-up of germ cell tumors, whereas precursor lesions do not secrete enough to make an early diagnosis possible [39,40,41].

Studies trying to identify prognostic factors for the development and progression of tumors have suggested a potential role of individual susceptibility, associated with SNP (single nucleotide polymorphisms) [42,43]. The identification of a combination of alleles that have the highest risk for potential malignant transformation is a promising screening approach in a distinct population [37,43].

At present, the gold standard for diagnosis remains histological analysis after biopsy [19].

For patients with female phenotype, there is limited proof for usefulness of gonadal biopsies, as the risk of developing gonadal malignancies is high and there is less reluctance to perform gonadectomy in these patients [44]. Gonadal biopsy has limited reliability, as tumors can be easily missed due to the diversity of cells and therefore might not securely exclude smaller tumors.

In order to ensure the right diagnosis, molecular characterization should be performed in patients with XY-DSD. Genetic analysis at an early timepoint is vital to establish endocrinological and surgical therapy as soon as possible [45]. This fact also limits the power of our study, as karyotyping was performed in every patient, yet molecular characterization was done in only 15 patients (43%) in total. To date, molecular characterization of XY-DSD is generally performed only in a quarter to a third of patients. Besides karyotyping, Y-derived sequences should be evaluated with PCR in patients with clinical aspects of possible XY-DSD [8]. Molecular genetic analysis with new technologies, such as next generation sequencing, will increase the number of diagnoses, and therefore simplify the stratification to a risk profile. This might also detect multiple gene aberrations, which are difficult to reveal [46,47].

But even if the correct diagnosis has been stated, no data exist on the exact risk for the development of germ cell tumors for the different entities. Considerations have to be made for every patient depending on age and molecular diagnosis as to whether gonads should be removed or rather retained to conserve gonadal function at least during puberty.

One has to notice that the frequencies of gonadal cell tumors are based on patient groups that received a biopsy or gonadectomy. Therefore, the estimated number of undiagnosed asymptomatic tumors might be even higher.

If gonadectomy has been performed, hormonal replacement therapy has to be initiated. To date there are no special protocols for hormonal replacement for patients with XY-DSD after surgery. Therapeutical regimens are often derived from other entities, such as premature ovarian failure or hypogonadotropic hypogonadism. In our patient cohort, one patient was administered an oral contraceptive instead of transdermal hormonal replacement as she demanded it. To strengthen acceptance for gonadectomy and following hormonal replacement therapy, further studies are needed to evaluate long-term therapeutic effects, such as bone structure and well-being [48]. In their study with 26 patients with cAIS, Birnbaum et al. showed that testosterone might be also suitable for hormonal replacement therapy, especially in patients with reduced sexual functioning, but the decision for or against has to be made on an individual basis [49].

For those who refuse gonadectomy, Dohnert et al. recommend a biannual screening program which should include gonadal imaging by ultrasound or magnetic resonance imaging (MRI), tumor markers and evaluation of hormonal levels. However, the authors emphasize that no reliable parameters for premalignant changes are applicable to date. In cases with suspicious findings laparoscopic biopsy and possibly necessary gonadectomy have to be performed [15].

Cools et al. also emphasize the need for a systematic screening to monitor retained gonads in patients with cAIS. While they state that monitoring by ultrasound is a preferred option due to the feasibility and non-invasiveness of the examination and should be performed with the start of puberty, this method is not suitable for abdominally located gonad and MRI is more sensitive [37,50].

If the decision has been made to postpone gonadectomy until secondary sex characteristics are developed, or even longer after informed consent, monitoring should be performed in the meantime, being aware of the limitations mentioned above.

In conclusion, multi-center studies with larger cohorts of patients experiencing the various conditions are urgently needed to clarify the risk assessment and offer patients reliable data for shared decision making. The decision whether to perform gonadectomy and the time point of surgery should be made depending on the patient’s needs regarding quality of life and fertility concerns, also taking into consideration the DSD-subtype and patient’s age [15].

## 5. Conclusions

The risk of developing gonadal germ cell tumors in patients with XY-DSD is increased over their lifetime. Guidelines concerning the necessity of gonadectomy in XY-DSD to avoid potential malignant transformation are still lacking. The risk of developing malignant germ cell tumors from rudimentary gonads has not been investigated sufficiently to date, as it is mostly based on individual cases due to the rarity of the condition.

In our cohort, 40% of all patients who underwent gonadectomy in our department showed abnormal histopathological findings. Preventive gonadectomy in patients with XY-DSD has to be taken into consideration, depending on patients’ age and molecular diagnosis. Concerning the ideal point of time for gonadectomy for the individual entities, further clinical studies with a broader patient spectrum and larger numbers are needed.

## Figures and Tables

**Table 1 ijerph-18-05648-t001:** Patient characteristics.

Patient ID	Diagnosis	Molecular-Genetic Analysis	Gonadectomy Y/N	Age at Gonadectomy	Histology	Clinical Characteristics
1	cAIS	no	yes	17	benign	no axillary hair, sparse pubic hair, vaginal hypoplasia, uterus aplasia, inguinal gonads
2	5α-reductase deficiency	yes	yes	23	seminoma	normal external genitalia, uterus aplasia, inguinal gonads
3	cAIS	no	yes	10	benign	normal external genitalia, uterus aplasia
4	cAIS	no	Yes	20	benign (assumed)	no axillary hair, sparse pubic hair vaginal hypoplasia, uterus aplasia, inguinal gonads
5	cAIS	no	yes	childhood	benign (assumed)	clitoral hypertrophy, uterus aplasia
6	cAIS	no	yes	40	**seminoma, Leydig-cell-tumor**	vaginal hypoplasia, uterus aplasia
7	cAIS	no	yes	26	benign	vaginal hypoplasia, uterus aplasia
8	cAIS	no	yes	18	benign (assumed)	vaginal hypoplasia, uterus aplasia inguinal gonads
9	cAIS	no	yes	12	benign (assumed)	no axillary hair, sparse pubic hair, vaginal hypoplasia, uterus aplasia, inguinal gonads
10	homozygous LH-receptor-deficiency	no	yes	17	benign (assumed)	no axillary hair, sparse pubic hair, vaginal hypoplasia, inguinal gonads, uterus aplasia
11	cAIS	no	yes	1	benign (assumed)	no axillary, sparse pubic hair, uterus aplasia, inguinal gonads
12	17-β-hydroxy-steroiddehydrogenase-deficiency	no	yes	0	benign (assumed)	vaginal hypoplasia, uterus aplasia, inguinal gonads
13	cAIS	yes	yes	17	benign	vaginal hypoplasia, uterus aplasia, abdominal gonads
14	cAIS	yes	yes	19	**bilateral Sertoli-cell-adenoma**	no axillary, sparse pubic hair, vaginal hypoplasia, uterus aplasia, abdominal gonads
15	cAIS	no	yes	18	benign	vaginal hypoplasia, uterus aplasia, inguinal gonads
16	cAIS	no	no	/	/	vaginal hypoplasia, uterus aplasia, inguinal gonads
17	cAIS	no	yes	18	benign	sparse pubic hair, vaginal hypoplasia and uterus aplasia, abdominal gonads
18	cAIS	no	yes	6	benign (assumed)	vaginal hypoplasia, uterus aplasia, inguinal gonads
19	cAIS	no	yes	17	benign (assumed)	vaginal hypoplasia, uterus aplasia
20	pAIS	yes	yes	11	benign (assumed)	vaginal hypoplasia, uterus aplasia, inguinal gonads
21	cAIS	yes	no	/	/	no axillary hair, vaginal hypoplasia, uterus aplasia, inguinal gonads
22	cAIS	yes	no	/	/	no axillary hair, vaginal hypoplasia, uterus aplasia, abdominal gonads
23	pAiS	yes	yes	10	benign (assumed)	vaginal hypoplasia, uterus aplasia, inguinal gonads
24	cAIS	yes	no	/	benign (assumed)	vaginal hypoplasia, uterus aplasia, intraabdominal gonads
25	cAIS	yes	yes (right side)	6	benign	vaginal hypoplasia, uterus aplasia, inguinal gonads
26	pAIS	yes	yes	16	benign (assumed)	clitoral hypertrophy, vaginal hypoplasia, uterus aplasia, inguinal gonads
27	cAIS	yes	no	/	/	sparse pubic hair, vaginal hypoplasia, uterus aplasia, inguinal hernia
28	cAIS	yes	yes	6, 12	benign (assumed)	vaginal hypoplasia, uterus aplasia, inguinal gonads
29	cAIS	no	yes	20	**Sertoli-cell-adenoma (right)**	vaginal hypoplasia, uterus aplasia, inguinal gonads
30	XY-DSD	no	yes	0	benign (assumed)	clitoral hypertrophy, vaginal hypoplasia, uterus aplasia, inguinal gonads
31	cAIS	no	no	/	/	sparse pubic hair, vaginal hypoplasia, uterus aplasia, inguinal gonads
32	cAIS	yes	no	/	/	vaginal hypoplasia, uterus aplasia, intraabdominal gonads
33	cAIS	yes	no	/	/	uterus aplasia, inguinal gonads
34	pAIS	no	yes	14	benign (assumed)	clitoral hypertrophy, uterus aplasia, inguinal gonads
35	cAIS	yes	yes	26	benign	clitoral hypertrophy, vaginal hypoplasia, uterus aplasia, abdominal gonads

Benign (assumed): no anamnestic reported (pre)malignant lesions after external gonadectomy. In bold: abnormal histological findings.

**Table 2 ijerph-18-05648-t002:** Review of the literature.

Study n = Number of Patients	XY-DSD Diagnosis	Histological Diagnosis	Number of Germ Cell Tumor
Hannema et al. (2006) n = 44 [26]	cAIS	GCNIS	2/44 (4.6%)
Cools et al. (2006) n = 43 [27]	46 XY-DSD	GDB and/or Dysgerminoma	11/14 (78.6%)
Beaulieu et al. (2011) n = 30 [28]	GD	GCT (Dysgerminoma or Seminoma)	9/51 (17.7%)
Wünsch et al. (2012) n = 40 [29]	Complete GD	GBD and/or Dysgerminoma	3/8 (37.5%)
	cAIS		0/7 (0%)
	others		2/25 (8%)
Nakhal et al. (2013) n = 14 [19]	cAIS	GCNIS	2/14 (14.3%)
		Sex cord tumor	1/14 (7.1%)
Slowikowska-Hilczer et al. (2015) n = 94 [30]	Complete GD Partial GD	GDB	11/29 (37.9%)
		GCT	5/29 (17.2%)
		GDB	1/29 (3.5%)
		GCNIS	15/29 (51.7%)
		GCT	2/29 (6.9%)
Liu et al. (2014) n = 102 [31]	cAIS	GBD	9/30 (30.0%)
	pAIS	GBD	3/18 (16.7%)
	GD	GBD	3/33 (9.1%)
Huang et al. (2017) n = 292 [23]	Complete GD	GCT	21/59 (35.5%)
	Partial GD	GCT	5/90 (5.5%)
	AIS	GCT	15/113 (13.2%)

GCN: germ cell neoplasia. GCNIS: germ cell neoplasia in situ. GBD: gonadoblastoma. GCT: germ cell tumor. GD: gonadal dysgenesis. cAIS: complete androgen insensitivity syndrome. pAIS: partial androgen insensitivity syndrome. AIS: androgen insensitivity syndrome.

## Data Availability

The data presented in this study are available on request from the corresponding author.

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
