# Peer review of "Malignant Germ Cell Tumors and Their Precursor Gonadal Lesions in Patients with XY-DSD: A Case Series and Review of the Literature"

_ijerph, 2021, doi:10.3390/ijerph18115648_

Round 1

Reviewer 1 Report

This manuscript suffers from two main issues: 1) grammar/style, 2) organization. 

1) This manuscript has many small grammatical and style errors that make reading boring. These errors go from simple typos "... to me made...", to grammar "...should be then chosen...", to incomplete sentences. 
One example of the latter is: "Therapeutical regiments are often derived from other entities". Because there is no follow up sentence, this phrase does not make much sense. What entities? What do the authors mean by this? These sentences seem just placed there and seem disconnected. They leave the reader hanging, asking for examples or, at the very minimum, smoother transitions. This makes reading quite boring and takes a lot away from a manuscript that could be a lot more interesting. 

2) This manuscript seems like a collage of 2 different manuscripts. The first is the case series, the second is the review. Although the title states ".. A case series and review of the literature", it doesn't mean that the two things should be separated. The Discussion is just a review (with one paragraph mentioning the study), and, as mentioned above, it is a list of paragraphs referencing the literature but with little transition into an organic discussion.

The authors should rework the Discussion trying to make it more organic, with better transitions and really discuss their results in the context of the published literature. 

At the end of the Introduction, the authors state: "To date no management guidelines.... are available", and "In our study.....". Both things (a critical discussion of the need for guidelines and a description of how their research fits into the literature) are missing.

There is a lot of room for improvement, and I suggest the authors to take advantage of it.

Author Response

Dear Reviewer 1,

thank you for the critical review resulting in useful comments and suggestions.

Point-by-point answers to the comments

1) This manuscript has many small grammatical and style errors that make reading boring. These errors go from simple typos "... to me made...", to grammar "...should be then chosen...", to incomplete sentences. 
One example of the latter is: "Therapeutical regiments are often derived from other entities". Because there is no follow up sentence, this phrase does not make much sense. What entities? What do the authors mean by this? These sentences seem just placed there and seem disconnected. They leave the reader hanging, asking for examples or, at the very minimum, smoother transitions. This makes reading quite boring and takes a lot away from a manuscript that could be a lot more interesting. 

Answer: The manuscript was been edited and corrected by a native speaker

2) This manuscript seems like a collage of 2 different manuscripts. The first is the case series, the second is the review. Although the title states ".. A case series and review of the literature", it doesn't mean that the two things should be separated. The Discussion is just a review (with one paragraph mentioning the study), and, as mentioned above, it is a list of paragraphs referencing the literature but with little transition into an organic discussion.The authors should rework the Discussion trying to make it more organic, with better transitions and really discuss their results in the context of the published literature. 

Answer: The discussion was completey revised and shortened. A discussion of the results of our study was highlighted in the context of the published literature. Case series and review were linked.

3) At the end of the Introduction, the authors state: "To date no management guidelines.... are available", and "In our study.....". Both things (a critical discussion of the need for guidelines and a description of how their research fits into the literature) are missing.

Answer: The link between our study and prospective guidelines was emphasized in the introduction:

“To date, no management guidelines concerning the need and optimal point of time for gonadectomy in postpubertal individuals with XY-DSD are available. In our study we retrospectively analyzed patients treated for XY-DSD at our department and who partly underwent bilateral gonadectomy in order to create data that can possibly be integrated into guidelines for the treatment of XY-DSD.”

Reviewer 2 Report

1. “DSD affects 1 to 4500-5000 live births, mostly due to genetic defects during sexual differentiation [3].”

DSD affects 1 in 4500-5000 live births.

2. Table 2 is not mentioned anywhere in the text, it is only presented. It should be introduced in the discussion at some point.

3. "Of those patients who underwent surgical procedure at our hospital, two were diagnosed with unilateral seminoma (20 %), one with a bilateral and one with a unilateral Sertoli-Cell-Adenoma (20 %).”

4. I understood that these 4 patients are presented in the results sections. Maybe it would be clearer is the authors stated here that these 4 patients are the ones whose cases are being presented.

5. “In comparison to other disorders of sexual development, patients with 46, XY-DSD and complete or partial gonadal dysgenesis are at higher risk for the development of germ cell tumors as they contain more better developed tissue [36].”

I do not understand the last part of this sentence: as they contain more better developed tissue.

6. “Of the totally 51 gonads, 9 (18 %) revealed tumor tissue [28].”

Of the total 51 gonads…

7. “Considerations for every patient have to me made depending on age and molecular diagnosis whether gonads should be removed or rather retained to conserve gonadal function at least during puberty.”

…patient have to be made depending…

8. “To date there exist no serum markers, such as ß-HCG or alpha-Fetoprotein, that might detect lesions early [44].”

To date there are no serum marker…

9. “To date there exist no special protocols for hormonal replacement for patients with XY-DSD after gonadectomy.”

To date there are no special protocols…

10. “If the decision has been made to postpone gonadectomy until secondary sex characteristics are developed, monitoring should be performed in the meantime to exclude, respectively detect premalignant lesions, if possible.”

I do not understand this part: to exclude, respectively detect premalignant lesions, if possible.

11. The discussion is too long, presenting the results from a lot of other studies. This makes it confusing and without a clear focus. I would suggest mentioning just a few of them, maybe it would be interesting to keep the ones with different results and the ones that are relevant to your conclusions.

12. “The risk of developing gonadal germ cell tumors in patients with XY-DSD is increased over their live-time.”

… over their lifetime.

Author Response

Dear Reviewer 2,

thank you very much for the critical review and useful comments.

Point-by-point answers to the comments:

  1. “DSD affects 1 to 4500-5000 live births, mostly due to genetic defects during sexual differentiation [3].”

DSD affects 1 in 4500-5000 live births.

Answer: The manuscript was edited and corrected by a native speaker

  1. Table 2 is not mentioned anywhere in the text, it is only presented. It should be introduced in the discussion at some point.

Answer: Table 2 was deleted in the manuscript according to point 11.

  1. "Of those patients who underwent surgical procedure at our hospital, two were diagnosed with unilateral seminoma (20 %), one with a bilateral and one with a unilateral Sertoli-Cell-Adenoma (20 %).”

I understood that these 4 patients are presented in the results sections. Maybe it would be clearer is the authors stated here that these 4 patients are the ones whose cases are being presented.

Answer: The fact, that the 4 patients with pathological findings are described as cases was pointed out in the result section. “Of the patients who underwent a surgical procedure at our hospital, two were diagnosed with unilateral seminoma (20 %), one with a bilateral and one with a unilateral Sertoli-cell-adenoma (20 %). Their history is presented below.”

  1. “In comparison to other disorders of sexual development, patients with 46, XY-DSD and complete or partial gonadal dysgenesis are at higher risk for the development of germ cell tumors as they contain more better developed tissue [36].”

I do not understand the last part of this sentence: as they contain more better developed tissue.

 Answer: this sentence was removed

  1. “Of the totally 51 gonads, 9 (18 %) revealed tumor tissue [28].”

Of the total 51 gonads…

Answer: The manuscript was edited and corrected by a native speaker

  1. “Considerations for every patient have to me made depending on age and molecular diagnosis whether gonads should be removed or rather retained to conserve gonadal function at least during puberty.”

…patient have to be made depending…

 Answer: The manuscript was edited and corrected by a native speaker

  1. “To date there exist no serum markers, such as ß-HCG or alpha-Fetoprotein, that might detect lesions early [44].”

To date there are no serum marker…

Answer: The manuscript was edited and corrected by a native speaker

  1. “To date there exist no special protocols for hormonal replacement for patients with XY-DSD after gonadectomy.”

To date there are no special protocols…

Answer: The manuscript was edited and corrected by a native speaker

  1. “If the decision has been made to postpone gonadectomy until secondary sex characteristics are developed, monitoring should be performed in the meantime to exclude, respectively detect premalignant lesions, if possible.”

I do not understand this part: to exclude, respectively detect premalignant lesions, if possible.

Answer: This sentence was changed to: “If the decision has been made to postpone gonadectomy until secondary sex characteristics are developed or even longer after informed consent, monitoring should be performed in the meantime, being aware of the limitations mentioned above”.

  1. The discussion is too long, presenting the results from a lot of other studies. This makes it confusing and without a clear focus. I would suggest mentioning just a few of them, maybe it would be interesting to keep the ones with different results and the ones that are relevant to your conclusions.

 Answer: The discussion was completely revised and shortened. A selection of the most suitable literature was made.

  1. “The risk of developing gonadal germ cell tumors in patients with XY-DSD is increased over their live-time.”

… over their lifetime.

Answer: The manuscript was edited and corrected by a native speaker